# Hyperlink Hijacking:
# Exploiting Erroneous URL Links to Phantom Domains

## ABSTRACT

Web users often follow hyperlinks hastily, expecting them to be correctly programmed. However, it is possible those links contain typos or other mistakes. By discovering active but erroneous hyperlinks, a malicious actor can spoof a website or service, impersonating the expected content and phishing private information. In *typosquatting*, misspellings of common domains are registered to exploit errors when *users* mistype a web address. Yet, no prior research has been dedicated to situations where the linking errors of web *publishers* (i.e. developers and content contributors) propagate to users. We hypothesize that these *hijackable hyperlinks* exist in large quantities with the potential to generate substantial traffic. Analyzing large-scale crawls of the web using high-performance computing, we show the web currently contains active links to more than 572 000 dot-com domains that have never been registered, what we term *phantom domains*. Registering 51 of these, we see 88% of phantom domains exceeding the traffic of a control domain, with up to 10 times more visits. Our analysis shows that these links exist due to 17 common publisher error modes, with the phantom domains they point to free for anyone to purchase and exploit for under $20, representing a low barrier to entry for potential attackers.

## CCS CONCEPTS

• **Information systems** → *Markup languages*; *Traffic analysis*; • **Security and privacy** → **Systems security**.

## KEYWORDS

web, hijackable, hijacking, hyperlinks, links, phantom domains, domains, vulnerabilities, spoofing, phishing, crawling, Common Crawl, typosquatting

**ACM Reference Format:**
Anonymous Author(s). 2023. Hyperlink Hijacking: Exploiting Erroneous URL Links to Phantom Domains. In *Proceedings of The Web Conference 2024 (WWW '24)*. ACM, New York, NY, USA, 10 pages. https://doi.org/XXXXXXX.XXXXXXX

## 1 INTRODUCTION

While the number of users accessing the web continues to grow [11], web research by academics passed its zenith in 2010 [9]. A few years later, cybercriminals took up the task, investigating research gaps and funding their future work with record incomes

[38]. Frequently their *modi operandi* includes simple techniques that exploit human fallibility at scale. An example of such an exploit is *typosquatting*, where an attacker registers a domain name that is a mistyped variant of another popular domain name [1]. This capitalizes on an error made by the *user*, who being an imperfect human, may occasionally mistype a domain in their web browser's address bar. In some cases the mistyped domain they inadvertently access hosts a spoofed (i.e. fraudulently impersonated) website. Humans tend to assume a website is authentic if it visually appears as expected, proceeding oblivious to the threat. It is burdensome and unrealistic to inspect security certificates for every visited website – assuming one even knows how – making exploits of this kind common and effective.

Web *publishers* are imperfect humans too. They, comprised of *developers* who write the code of the web and *contributors* who write the content on pages, do not necessarily inspect every link they create. Yet, unlike with typosquatting, web publishers do not bare the brunt of their own errors – users do, by simply following those links from authentic domains and assuming they are pointing to the intended destination domain.

As coding errors are known to be widely prevalent [17], errors by web publishers are also expected to be widespread. Despite this, no in-depth research has been undertaken on how links on the web can be exploited to hijack web traffic. Nikiforakis et al. skirt the subject most closely, reporting on five mistyped domains for the purposes of investigating Javascript library inclusion vulnerabilities [25]. Being otherwise out of scope, the concept is not explored further, providing a gap for further investigation. The massive scale of the web suggests a far greater number of errors are available to be discovered and characterized. Thus we hypothesize that the web contains significantly more erroneous links than have been previously reported, and that these errors have the potential to generate substantial exploitable traffic. These exploits include but are not limited spoofing/phishing websites and code inclusion vulnerabilities.

Here we distinguish *vulnerable domains* from *phantom domains*. Vulnerable domains include lapsed domains, abandoned by their former owners and are thus contributing to *link rot*, where outdated, persistent links to the domain remain on the web [18]. These types of domains are sought after on the premise that the former owners may want them back in the future, or the domains will come with inherent value due to the promotion imparted upon them by prior owners. This area is well explored [19], and the "Expired Domains" industry exists to support it. Phantom domains, by contrast, are domains that have never been registered, can be registered, and already have inbound links, as seen in Figure 1. Links to such domains are considered to be errors, since a link to a domain that has not yet been registered would not be intentional, or at very least, would not be sensible. The links to phantom domains are

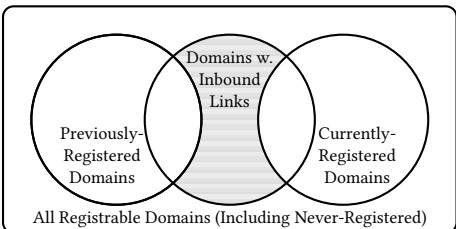

**Figure 1: Phantom domains in gray; includes never-registered domains with inbound links (i.e. hijackable hyperlinks)**

referred to a *hijackable*, since they can be seized and directed to an unintended destination.

This paper thus intends to estimate the number of phantom domain found on the web, characterize the hijackable links that lead to their existence and suggest mitigation and remediation strategies. By investigating the magnitude of the issue, as well as the means by which it occurs, we aspire to better understand the problem, raise awareness, reduce threat surface and ultimately thus reduce the risk to web users.

We begin by analyzing the Common Crawl dataset, which contains petabytes of crawler data covering the majority of the web since 2008 [2]. Using High-Performance Computing (HPC) resources at scale, we devise an approach to finding hijackable hyperlinks across the extent of the web. These links are then processed and filtered to remove links to active and vulnerable domains, leaving a shortlist of never-registered phantom domains. Once the list of phantom domains is compiled we inspect it for root causes of the errors leading to their creation, categorizing them into 17 common error modes such as missing or added characters, hyphenation errors, overflowing text and "fat finger" mistakes where adjacent keyboard keys are accidentally pressed. Some errors are not merely typos but instead poor practices, such as using placeholder domains in web page design templates. We investigate these errors in aggregate and in specific instances, using the findings to formulate mitigation and remediation strategies in consultation with experts.

The contributions of this paper are thus:

(1) The definition of hijackable hyperlinks and phantom domains, and a proposed framework for identifying them;
(2) Analysis of hundreds of billions of crawled web pages to determine the extent/severity of hijackable hyperlinks; and
(3) Proposal of mitigation/remediation countermeasures.

## 2 RELATED WORK

The related field of typo-squatting has been an area of interest to researchers since the early 2000s [1] with researchers aiming to map the landscape of typo-squatted domains [32], [35], [5], identify and classify variations of typo-squatting such as Combo-squatting [16], Bit-squatting [27], and Sound-squatting [24], and evaluate the impact of typo-squatting on users [15], [33] and companies [30].

Researchers have extensively studied the underlying reasons behind the susceptibility of certain domains to typo-squatting. Tahir et al. provides an overview of the factors influencing the mistyping of URLs including hand-anatomy, and keyboard layouts [37]. Similarly, Pochat et al. investigated the influence of international keyboard layouts on the mistyping of URLs and how this was being exploited [20]. These papers are highly relevant to the research presented in our paper as the factors influencing a user's mistyping of a URL is likely to also influence the mistyping of a URL by a developer.

Research into determining how these typo-squatted domains have been leveraged by attackers in order to make a profit has also been conducted. Alrwais et al. [4] provides an overview of domain parking services. These are services that resolve "parked domains" to sites containing large amounts of advertisements and split profits with the domain owner. Moore and Edelman [22] explored how typo-squatted domains are funded and found that 80% are supported by pay-per-click advertisements.

Sanchez-Rola et al. [29] conducted a study revealing that a significant majority of domains pose a risk to users by concealing the actual destination of links. Their research discovered that approximately 80% of websites mislead users by presenting incorrect `href` attributes on some of their links, with 45% of these links leading to domains different from what is displayed. This indicates that even if users meticulously inspect each link, there is no definitive way to determine their true destination. In a related study, Reynolds et al. [28] delve into the diverse interpretations of URLs by different parser implementations, which consequently introduce vulnerabilities that can be exploited. These variations in interpretation may cause certain clients to send requests to different hosts based on the same URL. Such inconsistencies in URL interpretation across systems can have serious security implications. Similarly, Kaleli et al. [13] explores situations in which the people generating content for social media make mistakes, namely the unintentional creation of URIs through grammar mistakes.

Nikiforakis et al. [25] examined web developers mistyping either the address or version of JavaScript libraries within their HTML or JavaScript code, allowing malicious actors to register the mistypes domain and compromise the script and site. They also explored the impact of developer mistyped URLs through traffic analysis of a common mistyping of `googlesindication.com`. They found that over 160,000 users visited the mistyped domain over 15 days. This work is comparable to ours but lists only 5 mistyped domains while focusing on JavaScript inclusion vulnerabilities. Here we seek to expand the scope of detection substantially, first determining the extent of the issue using analyses of large scale web crawls conducted using High-Performance Computing (HPC) resources. By doing so, we intend to define and detect hijackable hyperlinks across hundreds of billions of links spanning the web in order to understand their pervasiveness, evaluate how they are formed and develop countermeasures.

## 3 METHODOLOGY

In order to evaluate the extent and severity of hijackable hyperlinks found on the web, we devised methods which function at scale, evaluating the web en masse to output a list of phantom domains. We began by identifying a sufficiently large data source, which was processed over the course of months. From there, we developed methods to categorize the errors causing hijackable hyperlinks. Making observations from the resulting list of phantom domains, we then devised a method to investigate the sources of the errors.

Finally, we registered a selection of phantom domains and observed incoming traffic patterns. The methodology of each of these phases is discussed in this section.

## 3.1 Phantom Domains on the Web

Throughout this paper we reference hijackable hyperlinks and phantom domains closely. They are mutually dependent and cannot exist without each other – phantom domains by definition, require active but erroneous inbound links, and hijackable hyperlinks are those links.

More formally, server addresses on the the Internet can be represented by many possible domains names $D$, some portion of which are currently registered $D_r \subset D$, some which have been previously registered $D_p \subset D$ and some of which have never been registered $D_n \subset D$. An individual domain currently hosting a web site $d \in D_r$ provides source files in formats such as HTML, ASP, PHP, etc. The source files are rendered as web pages in users' browsers and may include hyperlink pointers $H_d$, which are links to destination domains. The destination domains referenced in $H_d$ can point to a valid, registered domains $h_r \in H_d$, a previously-registered domains $h_p \in H_d$ or a never-registered domains $h_n \in H_d$. A pointer $h_n$ is classified as a *hijackable hyperlink* and a domain to which hijackable hyperlinks may point $d_h \in D_n$ are defined as *phantom domains*.

In order to create a list of phantom domains we start by first evaluate the web for hijackable hyperlinks at scale. Related work thus far has largely depended on researchers operating customized web crawlers [14, 23, 26, 31, 34, 36]. This poses an inherent bottleneck in the detection of vulnerabilities on the web, as it is dependent on a small sample size, typically less than a million pages out of the many billions available, while being limited by infrastructure resources. We take a different approach, leveraging a thorough web crawler dataset complete with long-running historical records.

### 3.1.1 Data Sources.
The Common Crawl project has conducted crawls of the web since 2008 on a roughly monthly basis, with their dataset totaling petabytes in size [2]. In conjunction with the HPC batch processing resources of a major university, we analyzed their September 2022 to February 2023 domain level web graph. It distills billions of web pages into 88 million domain nodes with 1.68 billion edges. The edges of this graph represent hyperlinks found on crawled web pages. This dataset contains both domains crawled by Common Crawl and the destination domains linked to by the crawled pages, whether or not the destinations are valid or registered. As the crawled pages themselves must be registered and active at the time of the crawl, and therefore cannot be hosted on phantom domains, our analysis focused not on the pages that were crawled, but instead the destination domains found in the crawled pages' hyperlinks – approximately 52% of the graphs nodes. This includes valid pages with no outbound links which must be differentiated from phantom domains. In addition to Common Crawl's web graph, we also processed 104 of their prior crawl datasets. These datasets range in size, typically a few hundred compressed terabytes in size, each containing crawl data of a few billion pages per crawl [2]. Their data is formatted in Web ARChive (WARC) format, as well as processed files which contain information extracted from the raw WARC files.

The Internet Corporation for Assigned Names and Numbers (ICANN)'s Centralized Zone Data Service (CZDS) allows interested parties to access the zone files of the Top-Level Domain (TLD) (e.g. `.com`, `.org`, `.net`, etc.) [10]. Zone files contain all of the presently registered domains with corresponding server IP addresses in the TLD. In addition to the Common Crawl dataset, we requested and were granted a copy of ICANN's `.com` zone file via their CZDS. Notably, we chose to focus on the `.com` TLD since it by far the most dominant TLD with 8× the registered domains of the next biggest TLD and 72% more registered domains than the next 9 most popular TLDs combined [3]. The `.com` zone file used in early 2023 contained approximately 44 million domains.

We also query the Domain Name System (DNS), a distributed database linking domain names with the IP addresses that serve their contents. If a domain is available in the DNS system, it implies the domain is, or has been, registered.

Finally, we made use of the Internet Archive's *Wayback Machine* which contains archived versions of websites crawled since their inception in 1996 and totaling over 840 billion individual pages [12]. The dataset is available via API queries rather than a downloadable dataset, and as such, is a rate-limited resource to be used sparingly.

### 3.1.2 Filter destination domains from crawl dataset.
Identifying phantom domain within the Common Crawl's dataset was achieved through the discovery of all hyperlinks in the crawl data, followed by removal of destination domains identified as having been active or registered in the past. Links found by the Common Crawl's crawler, but with no trace of having been registered in the past, suggest the link is erroneous and thus hijackable, with the destination being a phantom domain. Identifying domains that were active in the past was achieved through the application of filtering stages. The ordering of the stages is primarily contingent on the efficacy of processing at scale – earlier stages operate on large data volumes quickly, later stages refine smaller data volumes more slowly. Intuitively, the reverse approach would be prohibitively time consuming and inefficient. The steps undertaken to identify hijackable domains are described below as well as outlined in Figure 2.

(A) *Extract targeted TLDs.* This research focuses on phantom domains in the `.com` TLD. As such, the Common Crawl web graph introduced in Section 3.1.1 was processed to extract only hyperlinks (i.e. network graph edges) pointing to `.com` destination domains. Additionally, Internationalized Domains Names (IDNs) – domains in non-Latin scripts or alphabets – were also excluded for processing simplicity, however their ASCII representations were not.

(B) *Remove domains found in prior crawls.* The Common Crawl project conducted 104 crawls prior to the research undertaken here. Any domains from step (A) that were crawled in those prior crawls was filtered out in this step, as finding crawled pages on those domains would suggest the domain was registered in the past and thus could not be a phantom domain. In order to accomplish this extremely computationally-heavy task, the Common Crawl WAT files – web archive files with computed page metadata, including hyperlinks – for each prior crawl were downloaded and each domain found in the `WARC-Target-URI` (i.e. crawled page) fields of the crawl's dataset was recorded. The resultant list of crawled domains was then used to generate Bloom filters. Bloom filters enable very fast set membership queries in constant `O(k)` time where k

**Figure 2: Filtering pipeline for extracting hijackable hyperlinks and phantom domains from Common Crawl dataset**

is the number of hash functions, enabling a means by which to quickly and accurately discard any domain that had been previously crawled from the web graph domain list. As Bloom filters can not have false negatives [6] it is ensured that phantom domains were incorrectly filtered out at this stage.

(C) *Remove domains that appear in the zone file.* Using the .com domain zone file provided by ICANN, domains that are presently registered and have a corresponding name server were filtered out of our dataset. Bloom filters were again used to query if a domain from step (B) was found in the zone file.

(D) *Remove domains that have a DNS record.* Through utilization of the dnspython toolkit [8], domains with a current DNS record were removed from the dataset. This was done through the use of the DNS resolver function.

(E) *Remove domains that have Wayback Machine history.* The remaining domains were queried via the Internet Archive's Wayback Machine API. Any domains with a historical crawl record were discarded from the dataset. This allowed remaining domains that were registered and active in the past but have since become inactive to be filtered. This step aids in catching domains that may be vulnerable due to link rot [18].

(F) *Final list of phantom domains.* The remaining filtered list is comprised of phantom domains – never-registered domains with active, hijackable inbound links.

## 3.2 Error Characterization

Once the dataset was filtered and the phantom domains were identified, the remaining domains were analyzed. The analysis sought to identify both the mistakes leading to the creation of a hijackable hyperlink and the Likely Intended Domain (LID).

*3.2.1 Determining the LID.* If a domain on a web page is the result of a mistake made by a developer, it is likely that the domain they typed is similar to the domain they intended to type [34]. The domain that a developer likely intended to use is referred to as the LID. Levenshtein distance was used to determine the LID for each phantom domain. Levenshtein distance [21] is a metric for measuring the difference between two strings. It can be used to determine the number of insertions, deletions and substitutions needed to convert one string into another. For example if a developer while attempting to type the domain example.com typed exampke.com a Levenshtein distance of 1 exists between the domains due to 1 substitution. In this example, this would indicate that example.com is the LID. For each of the phantom domain identified, the LID was determined through measurement of the Levenshtein distance between itself and each registered domain found in the zone file.

The LID was determined through the following steps:

(1) *Check zone file for similar domain(s).* Levenshtein distance was calculated between each phantom domain and each domain found in the .com zone file. This enabled finding domains that are the closest registered domain to the phantom domain.

(2) *Computationally categorize mistakes.* Once the LID was determined, the following substeps were taken to classify the type of error into the categories that will be outlined in Section 3.2.2: (i) Check if the domain contains any invalid characters. (ii) Using Python's difflib module, determine the additions and/or subtractions that need to be made to convert the phantom domain into the zone file domain. (iii) Check if the only character needed to be added or subtracted is a '-' which is indicative of a hyphenation error. (iv) Check if only an 's' is needed to be added or subtracted which indicates that the error was a pluralization error. (v) Check if mistake is a permutation error – two adjacent characters in the wrong order. (vi) Check if the difference between the two strings is that the phantom domain contains 'www' or the LID contains 'com'. (vii) Check if an incorrectly pressed key is a "fat finger" mistake using simple custom code which looks up adjacent keyboard letters. For QWERTY, AZERTY, and QWERTZ keyboards, key layouts determine the physical distance between the pressed key and the key that should have been pressed.

(3) *Manually spot check remaining phantom domains for placeholders.* Some phantom domain defy computational categorization and are most efficiently classified by through semantic understanding of context. Since the human brain remains the most efficient tool for this task and the data volumes were low due to earlier filtering stages, this final stage was done manually. These remaining phantom domains notably include placeholders domains included by developers as a poor coding practice. Ostensibly, the purpose of the placeholder domain is to be substituted by a valid domain when one is obtained.

Once the LID is determined, the phantom domain was classified according to the type of error.

*3.2.2 Types of Errors.* In order to determine the cause of the errors leading to the hijackable hyperlink, the mistake or typo made by the developer needed to be classified. By investigating the phantom domains, several trends were identified, necessitating a taxonomy for analysis. 17 error modes were identified in the dataset as seen in Table 1.

A complexity is implied in this analysis: some errors meet the description of multiple categories. For example, adding a hyphen is both a "Hyphenation Error" as well as a "Character Addition" error. These overlapping classifications are addressed through prioritizing specific errors over general ones. In the case of the given example, a "Hyphenation" error is more specific than a "Character Addition" error. The error categories listed above are sorted by specificity, and any reported errors populate the first (and thus most specific) error category that meets the error's characteristics.

*3.2.3 Ranking Phantom Domains.* When analyzing the large lists of phantom domains it was evident that some phantom domain are more valuable than others. As the presumed goal of a malicious actor would be to maximize inbound traffic, a method for ranking

**Table 1: Error categorization order – populating top down in first appropriate category**

| Error | Note or Example | Color |
|---|---|---|
| Invalid format | Unresolvable, e.g. `example_site.com` | |
| 1. IDN | `dømi.fo (xn–dmi–0na.fo)` | |
| 2. Add Hyphen | `example.com → ex-ample.com` | |
| 3. Remove Hyphen | `ex-ample.com → example.com` | |
| 4. Fat Finger | `example.com → rxample.com` | |
| 5. Permutation | `example.com → exapmle.com` | |
| 6. Remove Character | `example.com → exmple.com` | |
| 7. Add Duplication | `example.com → exaample.com` | |
| 8. Remove Duplication | `examplee.com → example.com` | |
| 9. Add Character | `example.com → exsample.com` | |
| 10. Overflow | `example.com → wwwexample.com` | |
| 11. Underflow | `example.com → examplecom.com` | |
| 12. Single Char Swap | `example.com → ecample.com` | |
| 13. Double Char Swap | `example.com → ecemple.com` | |
| 14. Remove Two Char | `example.com → exple.com` | |
| 15. Add Two Char | `example.com → exsamplye.com` | |
| 16. No Close Domain | Levenshtein > 5; no match includes placeholder domains, e.g. `yourmobiledomain.com` | |
| 17. Not Classified | Levenshtein ≤ 5, uncategorizable | |

the potential inbound traffic to the phantom domains was required. Several metrics were considered, such as the number of inbound hijackable hyperlinks and/or source domains. While this metric is logical, it could easily be spammed, particularly by Search Engine Optimization (SEO) companies who create large quantities of links that typically have low visibility. This approach also requires substantial computation. Harmonic centrality was also considered, particularly because this information is included in the Common Crawl web graph and thus readily available. Harmonic centrality is a measure of distance between a node and all other nodes in the graph. This approach is regarded as somewhat esoteric and was not used due to lack of ubiquity.

The selected approach was to rank phantom domains according to PageRank. PageRank was introduce by Brin and Page of Google in 1998 as a Markov chain for assessing the importance of a web page [7]. PageRank remains the de facto standard for ranking web pages, and this metric is provided in the Common Crawl web graph. For these reasons it was chosen for ranking the discovered phantom domains, noting that a PageRank can be calculated based on inbound links, regardless of if the domain is actually registered.

### 3.3 Sources of Hijackable Hyperlinks

Once the hijackable hyperlinks had been identified an additional processing step was undertaken to determine the source domains with pages linking to them. This was done through analysis of the Common Crawl's domain level web graph. The web graph offers an efficient means of determining all domains that point to a particular phantom domain.

The Common Crawl domain level graph includes all hyperlinked domains found in their crawls of the web, regardless of link validity or the destination domain's registration status, and thus includes phantom domains. By tracing back from the phantom domains along the edges using data analysis, the source domains which point to phantom domains were uncovered.

Finding the specific web page(s) on the domain with the offending hijackable hyperlinks was possible through two methods. The first was simply using search engine queries, which was effective for single instances but laborious at scale. The second is targeted crawling. If one knows a phantom domain and the source domain hosting the erroneous hijackable hyperlink, crawling the source domain for hyperlinks which point to the phantom domain uncovers the source of the hijackable hyperlink. This process was undertaken by finding false links on the offending domain using the Python `LinkChecker` module.

### 3.4 Traffic Analysis

To confirm our hypothesis that phantom domains have the potential to generate substantial exploitable traffic, it was necessary to evaluate traffic across a selection of phantom domains. In order to do so, the the three phantom domains with the highest PageRank in each of the 17 categories were identified, registered and connected to a hosting account serving a zero-length (i.e. blank) landing page.

The resulting web hosting logs provide data on incoming visitors. Although the page could be used to serve JavaScript-based tracking libraries to gather further information about the users we chose to take a less intrusive path, complete with ethics approval.

The information provided according to standard web logs included: (a) The IP address of the visitor; (b) The date and time of the visit; (c) The client's web request (e.g. HTTP GET/POST request); and (d) The visitor's *user agent*, which describes the software used to visit the page (e.g. Chrome/Firefox/Safari, or a crawler's name. Notably, the user agent is not verified and thus may be misrepresented by software or visitors. This is sometimes the case for crawlers which seek to disguise themselves by reporting a false user agent. The resultant data was analyzed to characterize the visitors over time and the results reported in Section 4.4.

## 4 RESULTS

In this section we report on the findings regarding the prevalence and characteristics of hijackable hyperlink across the web. Each of the Methodology subsections (3.1, 3.2, 3.3 and 3.4), has a corresponding Results section below.

### 4.1 Phantom Domains on the Web

Using the 88 million domains found in the Common Crawl webgraph, 43 020 911 `.com` domains were extracted. Figure 3 shows the number of domains that were filtered out of the dataset as described in Section 3.1. As shown, each stage filtered a large number of remaining domains, resulting in 572 126 `.com` phantom domain, representing approximately 1.3% of the `.com` domains analyzed.

### 4.2 Error Characterization

The methodology outlined in Section 3.2 was used to categorize the mistakes made by web publishers resulting in a hijackable hyperlink, with the errors shown visually in Figure 4 and outlined in full detail in Appendix Section A. Of the 572 126 phantom domains identified, 19% had a Levenshtein distance greater than five from any domain in the `.com` zone file. 27% were within a distance of 5 but could not be classified into any of the prior categories, and 1.2% were

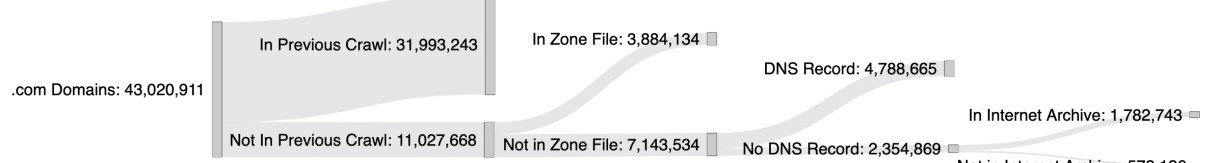

**Figure 3: Sankey of filtering process used to find phantom domains**

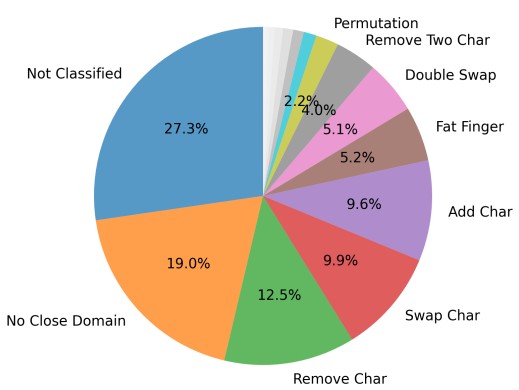

**Figure 4: Types of errors made by web publishers**

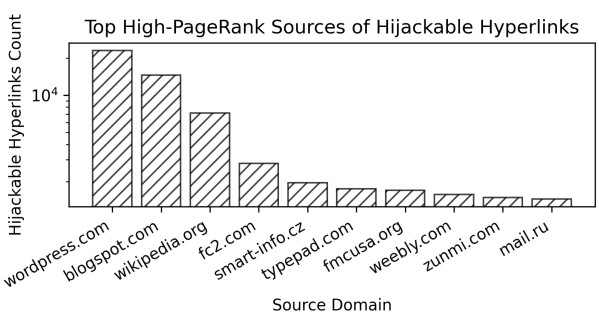

**Figure 5: Top 10 Sources of hijackable hyperlinks**

IDNs which are not considered errors per se, but for which error categorization was not undertaken.

### 4.3 Sources of Hijackable Hyperlinks

Through analysis of the source domains on which hijackable hyperlinks are found we can establish which web sites are more likely to contain this type of vulnerability. Figure 5 shows the relative frequency of hijackable hyperlinks on sites with the largest PageRank. These domains were observed to contribute a disproportionate number of hijackable hyperlinks. These are typically found alongside popular web Content Management System (CMS) such as Wordpress (`wordpress.com`), Blogger (`blogspot.com`) and FC2 (a blogging platform popular in Japan), among others. Wikipedia, with it's user-contributed content, as well a various webmail/content services, also contribute significantly. Notably, a religious organization produce a large number of hijackable hyperlink which appear to be long, randomly-generated string sequences of equal length, suggesting code-generated links. The prevalence of hijackable hyperlinks across these domains indicates that hijackable hyperlinks may be more likely to be generated by non-professional web developers or content contributors.

The vast majority of the hijackable hyperlinks found were on pages with low PageRank, often on domains running CMSs. In one observed instance, a General Data Protection Regulation (GDPR) cookie management extension was found to be automatically generating hijackable hyperlink in large quantities, the extension being employed on many independent web sites to comply with European Union privacy legislation.

Also present in the data is a large number of domains that are likely to be placeholders on templates but have been formatted in such a way that, while appearing to be a valid domains, are unregisterable. The most common example of this is domains containing an underscore, for example: `domain_goes_here.com`.

### 4.4 Traffic Analysis

Using the 51 phantom domains that were purchased, a plot of the cumulative traffic per day after registration was generated and is presented in Figure 6. The 3 phantom domains with the highest PageRank from each of the 17 error categories were purchased in September 2023. Additionally a random baseline/control domain was purchased and plotted for comparison.

The baseline control received no immediate hits, presumably due to being unfamiliar to crawlers who favor discovery via inbound links. Uniformly, the phantom domains were visited more quickly, presumably due to being discovered through existing hijackable hyperlinks. Traffic observed on the phantom domains exceeded that of the baseline upon registration. A week after registration, the baseline domain received more cumulative visits than 11.8% of the 51 domains registered. The other 88.2% exceeded the baseline. The median visits to the registered phantom domains amounted to 1445 ± 1262 inbound visits within the first week, with a maximum of 8481 visits after 7 days. For comparison, the baseline control domain received 821 visits after 7 days. Beyond the first week, the established trends continued unremarkably.

We observe that PageRank may not necessary provide an indicator of which phantom domains result in the highest immediate traffic, as shown in Figure 7. This is evidenced by the single most visits being attributed to a phantom domain with a mediocre PageRank compared to the other purchased domains. This domain is the anglicized version of an Asian-language phrase containing a double character swap. While PageRank is an effective metric for gauging

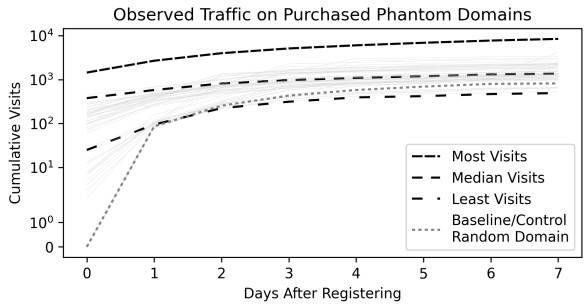

**Figure 6: Visitors to top 3 PageRanked phantom domains from each error category within the first week of registering**

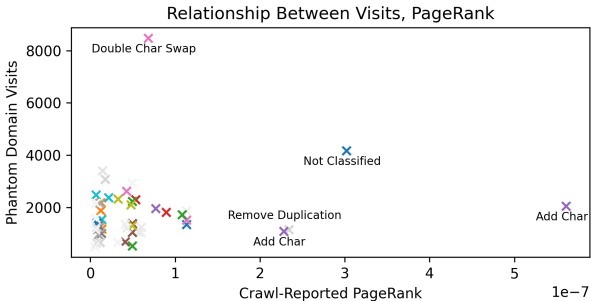

**Figure 7: First week of visits, colorized by error category**

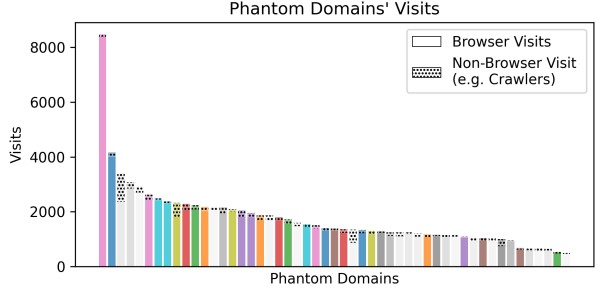

**Figure 8: First week of visits, showing browser and non-browser traffic, colorized by error category**

general traffic, other factors may be better predictors of phantom domain traffic. Similarly, error category does not provide an indicator of traffic as shown in Figure 8. Since errors appear not to correlate with visits, we presume that errors, being unintentional, are also distributed evenly across phantom domains with varying popularity popularity.

## 5 COUNTERMEASURES

As shown, hijackable hyperlinks exist in large quantities on the web, presenting a commensurate opportunity for exploitation. This raises the question about how the risk of hijackable hyperlinks can be addressed. Some countermeasures are presented here on

the basis of observations. Mitigation and remediation options are presented here, each divided into internal and external actions.

### 5.1 Mitigation

Mitigation refers to the lessening of an already-present threat. Given that the risk of hijackable hyperlinks has been shown to exist, mitigation will be discussed first.

#### 5.1.1 Internal Action.

**Improved coding and deployment practices.** The community of web developers have an opportunity to improve coding practices, both in their own linking and in the evaluation of the contributors-generated links. This may involve tests of links before deployment, log monitoring, and analysis of existing links.

**Inclusion in cybersecurity audits.** Large companies also often engage in cybersecurtiy audits and consultation. Based on the the results of this study, these firms are likely not considering the effect of phantom domains on their clients. Their clients may be either the source of hijackable hyperlinks or have phantom domains that are variations of their clients' brand(s). In either case, cybersecurity firms should begin auditing for phantom domains and the companies that hire them should demand this service.

**Brand protection.** Large companies often employ brand protection firms to purchase variations of their trademarks. From the work done here, many variations of brand names are available as phantom domains. Through WHOIS lookups it can be seen that in several instances these variations were missed by brand management firms. We avoid citing specific examples for safety and to avoid incrimination. We suggest brand management firms should employ the techniques outlined to monitor for phantom domains and the companies retaining them should demand this service.

#### 5.1.2 External Action.

**Detection by registrars.** Domain name registrars – companies who sell domains – often do various validations on purchases. Examples may be if buying a domain with a particular attribute, requiring the purchaser to talk with a customer service representative first. Similar checks can be implemented using the techniques outlined in Section 3, and employed when someone intends to register a phantom domain. Using a similar mechanism, purchase of phantom domains could be monitored or reported if subsequent phishing content is posted on the domain.

**Action by TLD administrators.** TLD and country code Top-Level Domain (ccTLD) (for specific countries, e.g. .ca, .cn, .ch, etc.) are empowered to enact policies for the TLDs under their administration. Through their policies, existing domains in the TLD can be required to ensure they do not direct to phantom domains, potentially freezing the domain until the hijackable hyperlinks are remedied. This admittedly is a heavy-handed approach, but may be applicable in some newer domain zones, such as those owned exclusively by companies.

**Inclusion in vulnerability databases.** Vulnerability databases exist to aid in auditing and cybersecurity tasks. Listing hijackable hyperlinks as a form of vulnerability in these databases would enable codebases to be audited for code that could potentially enable the creation of phantom domains.

**Name and shame.** As with vulnerability databases, a public website reporting offending domains with hijackable hyperlinks

would serve to encourage mitigation of phantom domains. A score representing the degree of risk posed by the domain hosting the hijackable hyperlinks could be generated, such as considering the number of hijackable hyperlinks and the PageRank of the source. These scores could be compiled on a public website, with the worst offenders at the top of a sorted list.

**Browser integration.** Browsers now typically warn users if they are visiting unsafe sites, giving them the chance to visit the site anyway if they see fit. Similar safety mechanisms may be possible either through direct integration into browsers, or via extensions/add-ons. This functionality can be an extension of other services like Google's SafeBrowsing service, which provides a safety check backend for Chrome and other browsers, yet other similar services also do exist from other vendors.

**Integration into web developer tools.** Tools exist for improving the quality of web pages. An example is Google Lighthouse, which in the past has incorporated vulnerability databases such as Snyk into their functionality. Tools of this sort often conduct automated audits appropriate to detect hijackable hyperlinks.

**CMS checks.** CMS sites are among the most common sources of hijackable hyperlinks. Ensuring that placeholder links used in page design templates are unregisterable is good practice and simple to confirm by CMS software. Additionally, CMSs often allow user-contributed add-ons/plug-ins/extensions/etc. which themselves have been observed to generate phantom domains on web pages. Since CMSs often depend on independent parties keeping their software up-to-date, it would be wise for CMS producers to include auditing functionality for hijackable hyperlinks in their software.

**Raising awareness.** Perhaps the most effective means of mitigation is simply raising awareness. This paper seeks to provide that function and welcomes efforts by others to do the same.

## 5.2 Remediation

Remediation, from the Latin for "cure", refers to the removal of existing threats. Due to the largely ungoverned nature of the web, widespread remediation efforts are inherently challenging. Some approaches to achieving remediation are presented here nevertheless.

### 5.2.1 Internal Action.

**Corrective action by web developers.** Web developers hold the most power to remediate existing hijackable hyperlinks on the domains they manage. Since hijackable hyperlinks are a subset of broken links – provided a phantom domain has not yet been registered by an attacker – finding and fixing broken links is an obvious recommendation. However, doing so gives few insights into the coding practice of the developers managing a domain to their supervisors. By employing Algorithm 1 those responsible for web sites can both remediate hijackable hyperlinks on their site(s), and be aware of failures in coding practices on behalf of their web developers.

The algorithm presented here presumes a relatively small number of invalid links, within the fair use of the Internet Archive's API. For greater quantities of hijackable hyperlinks, a filtering approach like the one used in this paper may be more appropriate.

### 5.2.2 External Action.

**TLD governance.** TLDs are empowered to enforce standards for

---

**Algorithm 1** Hijackable hyperlink detection for webmasters

```
1:  for page in web_site do
2:      links ← ExtractOutboundLinks(page)
3:      for link in page_links do
4:          if ResolveLink(link) is not resolveable then
5:              if CheckInternetArchive(link) has no record then
6:                  MarkAsHijackable(link)
7:              invalid_links ← link;
8:  GenerateInvalidLinksReport(invalid_links)
9:  SendReportToAdmin(admin_email, invalid_links)
```

---

the domains in their zone. In some cases, particularly for ccTLDs, enforcement of valid links can occur as a governance mechanism. Through enacting standards for domains, regardless of the degree to which these are enforced, can prevent future hijackable hyperlink or educate domain owners to their risks.

**TLD auditing.** By integrating checks for hijackable hyperlinks in order to ensure governance rules are being met, domain owners can be encouraged to remedy hijackable hyperlinks found on their domains. Auditing practices are often already in place at ccTLDs.

**Crawling services.** The Common Crawl and the Internet Archive already regularly crawl the web and, by extension, are well positioned to detect hijackable hyperlinks. By using these services to issue reports or notify webmasters, existing hijackable hyperlinks can be removed.

**Notifying webmasters.** When registering a domain, WHOIS and/or Registration Data Access Protocol (RDAP) records are generated containing the registration information of the domain. This information can be queried and used to automatically notify webmasters (via the domain's technical contact) when a hijackable hyperlink is discovered. This proves problematic for domains with many subdomains, but may be suitable for those operating a single website. This process would require a crawling service or other detection/notification service which engages in such activity.

**Proactive phantom domain purchasing.** If an entity becomes aware of phantom domains, it can work to identify the highest risk domains. By doing so, the entity can then purchase some or all of those domains to prevent them from being used as an attack vector.

**Raising awareness.** As with mitigation, perhaps the most effective means of mitigation is simply raising awareness. This paper seeks to provide that function and welcomes efforts by others to do the same.

## 6 CONCLUSION

As shown, hijackable hyperlinks exist in large quantities on the web with the potential to generate traffic to phantom domains well in excess of typical domains. These phantom domains in turn, can be used to host spoofed websites, phish private information from users, or deliver malicious code injections, to name but a few potential attacks. Through 17 error modes, hijackable hyperlinks are created by web publishers with no consistent safeguards in place. As such, we recommend a variety of mitigation and remediation strategies. Ultimately, awareness and improved practices are necessary to avoid the proliferation of hijackable hyperlinks and the risks they pose.

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

## A ERRORS BY CATEGORY

**Table 2: Domain error categorization – populating sequentially in first appropriate category**

| Error | % | Error | % |
|---|---|---|---|
| Invalid Format | 0.34% | 9. Add Char | 9.61% |
| 1. IDN (uncategorized) | 1.23% | 10. Overflow | 0.79% |
| 2. Add Hyphen | 0.15% | 11. Underflow | 0.02% |
| 3. Remove Hyphen | 1.02% | 12. Single Char Swap | 9.94% |
| 4. Fat Finger | 5.23% | 13. Double Char Swap | 5.11% |
| 5. Permutation | 2.18% | 14. Remove Two Char | 3.97% |
| 6. Remove Char | 12.53% | 15. Add Two Char | 0.60% |
| 7. Add Duplication | 0.01% | 16. No Close Domain | 19.00% |
| 8. Remove Duplic. | 0.96% | 17. Not Classified | 27.31% |

