# OpenReview forum: "Hyperlink Hijacking: Exploiting Erroneous URL Links to Phantom Domains"
_ACM.org/TheWebConf/2024/Conference — TheWebConf24 Oral_

### Official Review · Reviewer_BXka · 2023-11-17

**Novelty:** 3
**Technical Quality:** 3

**Review:**

This is a "new and old" problem. The fact that there are phantom hyperlinks has been known for quite some time (maybe since the late nineties). Of course, back then it was hard and expensive to register a domain. This reduced the incentive for attackers to take advantages in typos of domain names. There were also issues with homoglyphs in the DNS system that also got sorted out. However, this is a "new" problem at the same time because this problem does exist and is perhaps growing. Hence, this paper is an interesting study. It is also written very well and the evaluation is reasonably comprehensive.

However, my review scores do not reflect this optimism. Let me try to justify:

1. Even though the work is good, the fact is that the work is not that fresh (recall my new and old comment). Many would argue that these issues have been known for a long time and there is no additional value add.
2. Many would argue that the examples in Figure 5 are mostly user-generated content that are not expected to have high standards of authenticity and integrity anyway. If an unsuspecting user clicks a link, we can happily say that the fault is hers. This is because these are not very popular sites like bbc.co.uk or nytimes.com that engender a high degree of trust.

I still feel that a lot of constructive things can be done to improve the paper. Let me list them out.

1. Find issues in major sites like news portals, sports channels, technical sites, etc. These should be sites that you don't expect to have any hyperlink error. The authors have used the Common Crawl webgraph but they actually found problems in sites where users put in their content. In my view, Tranco should have been used.
2. Also, web crawlers (like the one Google uses) would be seeing such issues routinely. How do they deal with the problem? Do they use some kind of a reputation score?
3. There needs to be a deeper explanation about why these things happen from a human psychology or a web page design point of view?
4. Something needs to be proposed to create a secure site that the user can trust. Maybe the browser can show some icon in the corner that essentially says that all the hyperlinks in the page are genuine. Some such plugin should be created.

**Questions:**

1. Have major websites been analyzed for such problems? Let us take out user generated content from the equation.
2. Have CS psychologists looked at this issue? Do more errors happen let's say at night or with overworked developers?

**Ethics Review Description:**

This is fine as per me.

**Reviewer Confidence:**

3: The reviewer is confident but not certain that the evaluation is correct

**Scope:**

3: The work is somewhat relevant to the Web and to the track, and is of narrow interest to a sub-community

---

### Official Review · Reviewer_28YZ · 2023-11-22

**Novelty:** 6
**Technical Quality:** 6

**Review:**

The paper measures the prevalence of domains hyperlinked from other websites but never registered.

I find the paper to cover a very interesting security flaw, attributable to human oversight when inserting new domains, and a novel angle compared to prior work on vulnerable domains such as expired domains. Of course, the paper is limited to measuring how many _potentially_ exploitable domains are left out there. Inherently, the paper does not and cannot measure _actual_ exploitation of phantom domains, as by definition the domain would have been registered.

I find the measurement to be nicely executed. I find that the methodology section should still be improved quite significantly:
* an explicit reasoning why phantom domains are important could make it very clear why they are an issue (the web developer could never have intended to use such domains since they have never existed)
* 3.1: the attempt at a formal definition is not very clear, e.g., there is no formally expressed relation between `D` and `H`
* 3.1.2: how much work would it be to expand the analysis to other TLDs than `.com`?
* 3.2.1: the last step is a manual spot check. Can you elaborate on the process and the data volumes? I would appreciate more concrete numbers instead of a "low" data volume. What percentage of domains was spot checked?
* 3.2.2: the error categorization can be very valuable, as it could be reused as a taxonomy for future work, one that originates from actual real-world data observations. However, it would be useful to clarify exactly how this taxonomy was generated, i.e., was there a specific systematic procedure to identify the error types?
* 3.3: why was `LinkChecker` used to crawl links on source domains, instead of using the Common Crawl graph?

Regarding ethics, I found the description to be short but OK. The only concern I have, is that the IP address of the visitor is also considered PII. How was this handled in a privacy-preserving manner?

The analysis is correct, but the least well developed part in my opinion. Given that this would yield the most insights into this phenomenon, I would have expected a more elaborate and more accurate evaluation. For example, an analysis of the most common targets could have been interesting. It is also not clear to me whether the Traffic Analysis controls for the _source_ website in any way: I would expect that a phantom domain linked from a very popular website would receive a lot more organic traffic, without this having any other meaning. This is already touched upon by the vague statement that _"other factors may be better predictors of phantom
domain traffic"_ -- an exploration of what those other factors might be, would have been a valuable addition.
In general, the analysis just scratches the surface, but does not yield deeper insights into the reasons behind these errors and more interesting patterns.

I appreciate the extensive list of proposed countermeasures, which goes beyond purely technical measures.

I expect more discussion of the limitations of the approach, such as the selection of data sets. For example, historical zone files or WHOIS data might have useful to identify additional already-registered domains. A check on whether phantom domains are actually registerable (cf. reference [20]) could also be useful in case restrictions on domain registration apply. As for the analysis, in 4.4, passive DNS data could have been used to quantify traffic levels more comprehensively. Which other data sources would have been useful to filter hyperlinks or gain more insights? Was there a specific reasoning for the selection of data sources?

Personally, the mention of High-Performance Computing resources does not particularly impress me.

Smaller comments:
- S1: _"These links are then processed and
filtered to remove links to **active and vulnerable** domains, leaving
a shortlist of never-registered phantom domains."_ -- is this "links to active domains and links to vulnerable domains" or "links to domains that are both active and vulnerable"?
- S4.3: regarding the GDPR cookie management extension, did it generate the exact same hijackable hyperlink on every website where it was embedded, or a different one for each website?
- I would replace Figure 4 by Table 2; the pie chart obfuscates the smallest classes and otherwise contains the exact same data but at less precision. I would suggest to then sort Table 2 by percentage.
- Printing Algorithm 1 feels unnecessary for me, it just repeats the scanning method. This space could be saved for more analysis.

In summary, the problem is very interesting, and the paper already makes good progress towards understanding it better, but some flaws remain in (the description of) the methods and the analysis.

**I have read the rebuttal.**

**Questions:**

- Can you elaborate on the manual spot check process and the data volumes?
- Was there a specific systematic procedure to identify the error types?
- Why was `LinkChecker` used to crawl links on source domains, instead of using the Common Crawl graph?
- Which other data sources would have been useful to filter hyperlinks or gain more insights? Was there a specific reasoning for the selection of data sources?

**Reviewer Confidence:**

3: The reviewer is confident but not certain that the evaluation is correct

**Scope:**

4: The work is relevant to the Web and to the track, and is of broad interest to the community

---

### Official Review · Reviewer_uAHM · 2023-11-24

**Novelty:** 6
**Technical Quality:** 4

**Review:**

The authors explore the threat created by content creators and developers making errors in URLs they include in webpages. The authors specifically focus on errors in the domain name part of the URL - phantom domains - including various typing errors and leaving placeholder domains in URLs.

The submission has the potential to be a very nice research paper if some issues are fixed. It might be possible to fix these issues for the final submission.

To find phantom domains, the authors search large sources of web content (Common Crawl) and exclude registered and expired domains. To achieve this, they use several sources, including a zone file (they focus on .com domains only), active DNS, Wayback Machine, and Common Crawl. While I admire the authors' effort in filtering, I am not satisfied with their approach for two reasons. First, this filtering is incomplete. The authors analyze roughly six months of data containing 44m .com domains. For filtering, however, they only used one snapshot of the .com zone file even though it has a daily fluctuation of tens of thousands of domains, potentially missing millions of domains in this timeframe. These domains missed from filtering do not represent phantom domains of erroneous URLs, leading to a potential overestimation in the paper. Other data sources used for filtering in the paper are not as complete as zone files, and their effectiveness was not analyzed in this work, leaving us wondering how accurate the estimation is. Another additional data source that could have been used is passive DNS. I would condition the papers' acceptance on improving the filtering process.

An essential missing information is the exact number (or percentage) of unregistrable domains among the final set of phantom domains, as attackers cannot exploit those.

On traffic analysis, it seems like phantom domains do not receive a lot of visitors. I would have loved to see an estimate on the number of unique visits (e.g., based on source IP). A pDNS source could have been used to estimate traffic to phantom domains not registered by the authors.

An important research question not explored in the paper is whether there are phantom domains already exploited in the wild. How do we know that an attacker did not register some of the domains after an erroneous link was published?

This is the first academic work focusing on the problem of erroneous hyperlinks in general, even though there has been previous work related to the topic. The authors claim [25] is closest to their work; however, [13] addresses a subset of the same problem the authors explore in this submission. Also related is work by Szurdi et al. on "Email Typosquatting," where authors discovered job advertisements with mistyped email addresses, resulting in users sending their CVs to typosquatters.

The paper is well written. There are a few duplicate/erroneous references in the bibliography, like 13-14, 22-23, and 25-26.

[13] To Err.Is Human: Characterizing the Threat of Unintended URLs in Social Media.
[25] You are what you include: large-scale evaluation of remote javascript inclusions.

**Questions:**

Could you fix the issues mentioned above? Would you be able to use more zone files for filtering (at least one at the beginning of the analysis and one at the end)? Could you add the number of unregistrable phantom domains? Could you better estimate the traffic phantom domains received?

What do you think about the abuse of phantom domains in the wild? It could be analyzed by comparing the registration date of the domain with the webpage crawl date (or using passive DNS traffic).

**Reviewer Confidence:**

4: The reviewer is certain that the evaluation is correct and very familiar with the relevant literature

**Scope:**

4: The work is relevant to the Web and to the track, and is of broad interest to the community

---

### Official Review · Reviewer_UURB · 2023-12-04

**Novelty:** 6
**Technical Quality:** 6

**Review:**

Thank you for submitting your work to WWW 2024. Overall, I liked your paper and it fits the conference venue well. I have only minor concerns about technical correctness that I hope you address in your revision/camera-ready version.

Considering how you filter phantom domains, I struggle to understand why you are not using WHOIS information? Specifically, it would easily allow you to tell whether a domain is registered or not (also for domains with WHOIS privacy or from GDPR regions). This would allow you to condense steps B-E (of Figure 2) into one that is also more accurate and less of an approximation. You can also take WHOIS history into account (e.g. via DomainTools). A brief discussion on why that is not possible or practical would be helpful.

Since you base your work on the Common Crawl dataset, which includes pages at various depths, I am wondering whether the phantom domains you find are actually used on homepages of websites, or whether they might be used on subpages that are not linked from the main websites anymore (but are linked from some other third-party website) and if they are actually orphaned? (see also the work by Pletinckx et al., https://doi.org/10.1145/3460120.3485367)

The paper would also benefit from a more explicit positioning of phantom domains in the space of of domain-name registering attacks and how it is different from domain name "use after free" attacks, that is, that phantom domains are distinct in that the domain was never used before (Figure 1 is a first step into this direction, but some more explicit labeling of the Figure with different attacks would help).

Finally, many of the mitigations you mention do not appear realistic or practical to me. Quite a few of them are heavy-handed (e.g., action by TLD administrators/registrars, etc.) and go against the spirit of ICANN and TLD management. Contrasting/characterizing the various mitigations by effort/likelihood to be adopted should help to paint a more realistic picture of what could happen.

**Questions:**

- How accurate is your filtering pipeline in terms of domains truly being phantom domains?
- Why did you limit yourself to .com domains?
- On what sites do phantom domains occur? Homepage or specific paths?

**Reviewer Confidence:**

4: The reviewer is certain that the evaluation is correct and very familiar with the relevant literature

**Scope:**

4: The work is relevant to the Web and to the track, and is of broad interest to the community

---

### Decision · Program_Chairs · 2024-01-22

**Decision:**

Accept (Oral)

**Comment:**

# Summary

 This paper studies the interesting problem of "phantom domains", which are domains that have hyperlinks to the domain but are not registered. The security concerns is that adversaries can register these domains and take advantage of the pre-existing hyperlinks (and the implicit trust that already exists). The paper presents a system to detect phantom domains, and provides an estimate on the likelihood of the impact of existing phantom domains.

 # Strengths

 + Interesting and novel potential attack vector.
 + Found several phantom domains where hyperlinks exist on important websites, likely due to programmer error.
 + Paper suggests several possible defense mechanisms.
 + Paper is be of wide relevance to TheWebConf audience.

 # Weaknesses

 - One reviewer questions the core novelty of the work (as from one perspective this is similar to the broken links issue or typosquatting).
 - Impact of phantom domains is unclear (perhaps attackers have already registered all the "good" phantom domains).

 # Recommendation

 While there is some disagreement among the reviewers on the outcome of this paper, many of the reviewers are in favor of accepting this paper. Some concerns are about the novelty of the paper, as the idea of "broken links" is as old as the web itself. Other concerns implicitly suggest exciting follow-on work that can be done in this area (e.g., identifying if attackers have already taken advantage of phantom domains). Overall, I believe that this paper investigates an interesting new possible attack vector and will likely spark follow-up work in this area. The authors also engaged in the discussion about this paper, and the changes that they suggested based on reviewer feedback will substantially improve the paper.

 ---